# Using WGCNA (weighted gene co-expression network analysis) to identify the hub genes of skin hair follicle development in fetus stage of Inner Mongolia cashmere goat

Zhihong Wu[1]☯, Erhan Hai[1]☯, Zhengyang Di[1], Rong Ma[1], Fangzheng Shang[1], Yu Wang[5], Min Wang[1], Lili Liang[1], Youjun Rong[1], Jianfeng Pan[1], Wenbin Wu[6], Rui Su[1], Zhiying Wang[1], Ruijun Wang[1], Yanjun Zhang[1,4]*, Jinquan Li[1,2,3,4]*

1 College of Animal Science, Inner Mongolia Agricultural University, Hohhot, Inner Mongolia, China, 2 Key Laboratory of Animal Genetics, Breeding and Reproduction, Hohhot, Inner Mongolia Autonomous Region, China, 3 Key Laboratory of Mutton Sheep Genetics and Breeding, Ministry of Agriculture, Hohhot, China, 4 Engineering Research Center for Goat Genetics and Breeding, Hohhot, Inner Mongolia Autonomous Region, China, 5 College of Veterinary Medicine, Inner Mongolia Agricultural University, Hohhot, Inner Mongolia, China, 6 Zhenlai Hehe Animal Husbandry Development Co., Ltd, Baicheng, China

☯ These authors contributed equally to this work.
* lijinquan_nd@126.com (JL); imauzyj@163.com, imauzyj@imau.edu.cn (YZ)

**Data Availability Statement:** All relevant data are within the manuscript and its Supporting Information files.

## Abstract

### Objective

Mature hair follicles represent an important stage of hair follicle development, which determines the stability of hair follicle structure and its ability to enter the hair cycle. Here, we used weighted gene co-expression network analysis (WGCNA) to identify hub genes of mature skin and hair follicles in Inner Mongolian cashmere goats.

### Methods

We used transcriptome sequencing data for the skin of Inner Mongolian cashmere goats from fetal days 45–135 days, and divided the co expressed genes into different modules by WGCNA. Characteristic values were used to screen out modules that were highly expressed in mature skin follicles. Module hub genes were then selected based on the correlation coefficients between the gene and module eigenvalue, gene connectivity, and Gene Ontology (GO)/Kyoto Encyclopedia of Genes and Genomes (KEGG) enrichment analysis. The results were confirmed by quantitative polymerase chain reaction (qPCR).

### Results

Ten modules were successfully defined, of which one, with a total of 3166 genes, was selected as a specific module through sample and gene expression pattern analyses. A total of 584 candidate hub genes in the module were screened by the correlation coefficients between the genes and module eigenvalue and gene connectivity. Finally, GO/KEGG functional enrichment analyses detected WNT10A as a key gene in the development and maturation of skin hair follicles in fetal Inner Mongolian cashmere goats. qPCR showed that the

**Funding:** This work was supported by National Natural Science Foundation of China in the form of a grant awarded to YZ (31860627) and Plan Project of Science and Technology in Inner Mongolia in the form of a grant awarded to YZ (2019GG243).

**Competing interests:** The authors have declared that no competing interests exist

expression trends of 13 genes from seven fetal skin samples were consistent with the sequencing results, indicating that the sequencing results were reliable.n

## Introduction

China has a long history of cashmere goat breeding, with >30 pure-breeding varieties, including Inner Mongolian, Liaoning, and Tibetan cashmere goats, as well as many improved hybrids. Cashmere goats thus play an important role in the development of China's animal husbandry, and China is the world's largest producer of cashmere fibers [1]. However, rapid developments in the cashmere textile industry mean that the requirements for cashmere fineness are becoming increasingly strict, with a consequent need for the industry to cultivate excellent cashmere goat varieties.

Cashmere goats produce wool from primary hair follicles and cashmere from secondary hair follicles. Most research on skin and hair follicles in cashmere goats has focused on changes in the primary and secondary hair follicles during their growth, degeneration, and resting stages, and on the mechanisms of their related genes. The fetal development of primary and secondary hair follicles has been less well studied, and studies of fetal skin and hair follicle development have largely considered the occurrence of primary and secondary hair follicles (fetal period 45–65 days). However, there have been no reports on the maintenance of hair follicle structure and skin hair follicle homeostasis by epithelial cells and fibroblasts during the development and maturation of the primary and secondary hair follicles (115–135 days), even though the stability of the epithelial cells and fibroblasts has been shown to be a key requirement for the hair follicle to enter the hair cycle and to maintain the normal structure of the hair follicle in other mammals [2–7].

The early stages of skin and hair follicle development in Inner Mongolian cashmere goats has been divided into seven important stages (45–135 days). Cashmere goat wool is produced by primary hair follicles, while cashmere is produced by secondary follicles. The primary hair follicle primordium has been formed in all areas in cashmere goats at 55–65 days, some primary hair follicles have matured to form complete hair follicle structures by 105 days, and primary hair follicle development is basically complete by 135 days (Fig 1A). The primordial body of secondary hair follicles can be observed on the temples and neck at 65 days of fetal development, after which secondary follicles begin to grow and develop in various parts of the body, have developed and formed by 105 days, and gradually become mature after 125 days (Fig 1B) [8].

Transcriptome sequencing can reflect the gene expression in a species at a specific time and in specific tissues. The occurrence and fetal development of skin and hair follicles in Inner Mongolian cashmere goats was examined in detail by transcriptome sequencing of the seven stages [9]; however, although some key genes related to secondary hair follicle development were identified, no genes highly related to the development and maturation of fetal skin follicles were reported.

Weighted gene co-expression network analysis (WGCNA) is a popular systems biology method that describes gene correlations based on microarray databases to detect highly correlated gene clusters (modules). By identifying the intrinsic or central genes in these modules, these co-expressed gene clusters can summarize the modules and can then be related to the phenotype. Further analysis can also identify key and central participants in the module, thus helping to identify possible therapeutic targets or candidate biomarkers [10, 11].

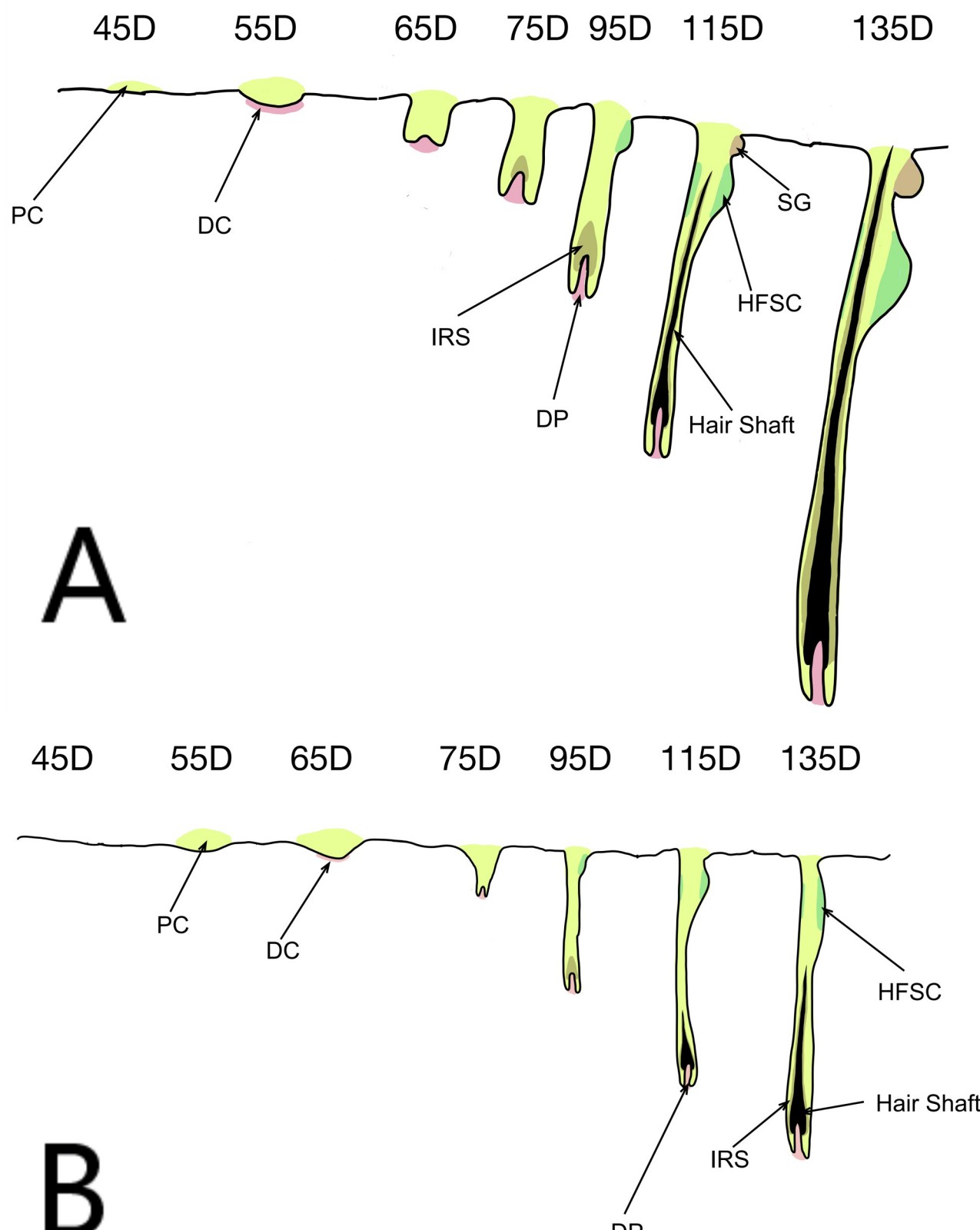

**Fig 1. Schematic diagram of skin hair follicle development.** (A) Primary hair follicles; (B) secondary hair follicles. PC: Placode; DC: Dermal condensate; DP: Dermal papilla; IRS: Inner root sheath; HFSC: Hair follicle stem cells; SG: Sebaceous gland.

## Materials and methods

### Ethics statement

Fetal skin samples were collected from Inner Mongolian cashmere goats in accordance with the International Guiding Principles for Biomedical Research Involving Animals. The procedure was approved by the Animal Ethics Committee of the Inner Mongolia Academy of Agriculture and Animal Husbandry Sciences, which is responsible for Animal Care and Use in the Inner Mongolia Autonomous Region of China. No specific permissions were required for these activities and the animals did not include any endangered or protected species.

### Skin sample preparation for quantitative real time-polymerase chain reaction (QPCR) validation

Samples were obtained from the Inner Mongolia Jinlai Animal Husbandry Technology Co., Ltd. (Hohhot, Inner Mongolia). The cashmere goat farm met the relevant Chinese national standard experimental animal environment and facilities requirements (gb14925-2010). According to the breeding records, we performed caesarean sections at different gestational ages. Surgery was carried out using general anesthesia and the ewes were monitored and nursed. Before the operation, the experimental animals were forbidden to eat and drink water. First, atropine (Quanyu, China) was injected intramuscularly with 0.04ml/kg, Half an hour later, xylazine hydrochloride injection (Shengxin, China) was injected intramuscularly with 0.1–0.15ml/kg. All operating personnel were qualified veterinarians with at least 3 years of clinical experience. We collected samples of fetal skin from three fetuses for each period (45, 55, 65, 75, 95, 115, and 135 days). The embryos were cleaned with diethyl pyrocarbonate water after cesarean section, and skin samples (1 cm$^2$) were collected quickly from the fetuses using sterile, enzyme-free disposable scalpels and forceps. The samples were then put into storage tubes, numbered, frozen rapidly in liquid nitrogen, and stored at -80˚C until use. The sampling process was conducted in strict accordance with animal welfare requirements.

### Data sources

This study used previous preliminary test results [9]. Fetal samples were collected as described above. The 21 sets of samples were subjected to RNA-Seq to obtain gene expression data for each gene sample. The gene expression data for each sample was normalized using the fragments per kilobase of exon per million fragments method. We have specified where the minimal data set underlying the results described in supporting information.

### Division of modules

Before WGCNA, we screened and filtered the selected gene set to remove low quality genes or samples with an unstable impact on the results to improve the accuracy of network construction. A weighted gene co-expression network was constructed using the WGCNA (v.147) package in R language [11]. The cor function (Pearson's correlation coefficient) was first used to calculate the correlation coefficient between any two genes, and the picks of threshold function was then used to obtain the appropriate soft threshold (power value with correlation coefficient reaching the plateau period or >0.8 was selected as the parameter for subsequent analysis), to construct an adjacency matrix in accordance with a scale-free network. The tomsimilarity function was used to transform the adjacency matrix into a topology overlap matrix, which was clustered using the hclust function (average), and the first principal component of each clustering module was calculated as the module eigenvalue (ME) using the

moduleeigengenes function. Modules with an eigenvalue similarity >0.85 were merged into one module, with a minimum of 50 genes in each module.

## Screening candidate hub genes

We used the ME to draw a sample expression pattern heat map for Inner Mongolian cashmere goat fetal skin and hair follicles in the seven periods, and selected the highest expression module in the mature stage as the candidate module. Important parameters of the genes in the module (correlation coefficient with ME and connectivity) were analyzed using the cor function (Pearson's correlation coefficient) to calculate the correlation coefficient between the gene and the characteristic value of each module (P value calculated by Student's $t$-test), and the total correlation coefficient of each gene with other genes was calculated in the adjacent matrix to determine the connectivity of each gene. The correlation coefficients between genes and the ME and gene connectivity identified candidate hub genes in the candidate modules.

## Enrichment analysis of key candidate genes with Gene Ontology (GO)/ Kyoto Encyclopedia of Genes and Genomes (KEGG) function

The candidate hub genes of specific target modules were subjected to KEGG pathway and GO functional enrichment analysis using cluego in Cytoscape 3.7.2 (http://apps.cytoscape.org/apps/cluego). KEGG pathways were analyzed using the KEGG database (http://www.KEGG.jp) and GO functional enrichment analysis was carried out using the GO database (http://geneontology.org/).

## QPCR verification

Total RNA was isolated from skin tissue using TRIzol reagent (Takara, China) and reverse-transcribed into cDNA using a Primer Script™ Reagent Kit (Takara, China) QPCR. QPCR was carried out on an ABI 7300 system (ABI, USA) using FastStart Universal SYBR Green Master according to the manufacturer's instructions. The thermal cycling conditions used for QPCR were 95°C for 5 min, followed by 45 cycles of 95°C for 30 s and 60°C for 45 s. Experiments were carried out using three biological and technical replicates, respectively. Data were expressed as mean±standard error.

# Results

## Construction of weighted gene co-expression network

The applicable power value for this test was four (Fig 2). Ten co-expression modules were created and coded as blue, black, green, turquoise, pure, red, brown, yellow, magenta, and pink modules, including 3,166, 305, 494, 10,378, 62, 439, 2,958, 1,305, 236, and 293 genes, respectively (total 19,636 genes) (Fig 3).

## Selection of specific target modules for skin hair follicle development and maturation in fetal cashmere goats

The sample expression pattern heat map (Fig 4) showed that genes in the blue module were most highly expressed in three samples at 115 days and three samples at 135 days, consistent with the developmental periods for skin and hair follicles in fetal Inner Mongolian cashmere goats. The blue module was accordingly selected for further study.

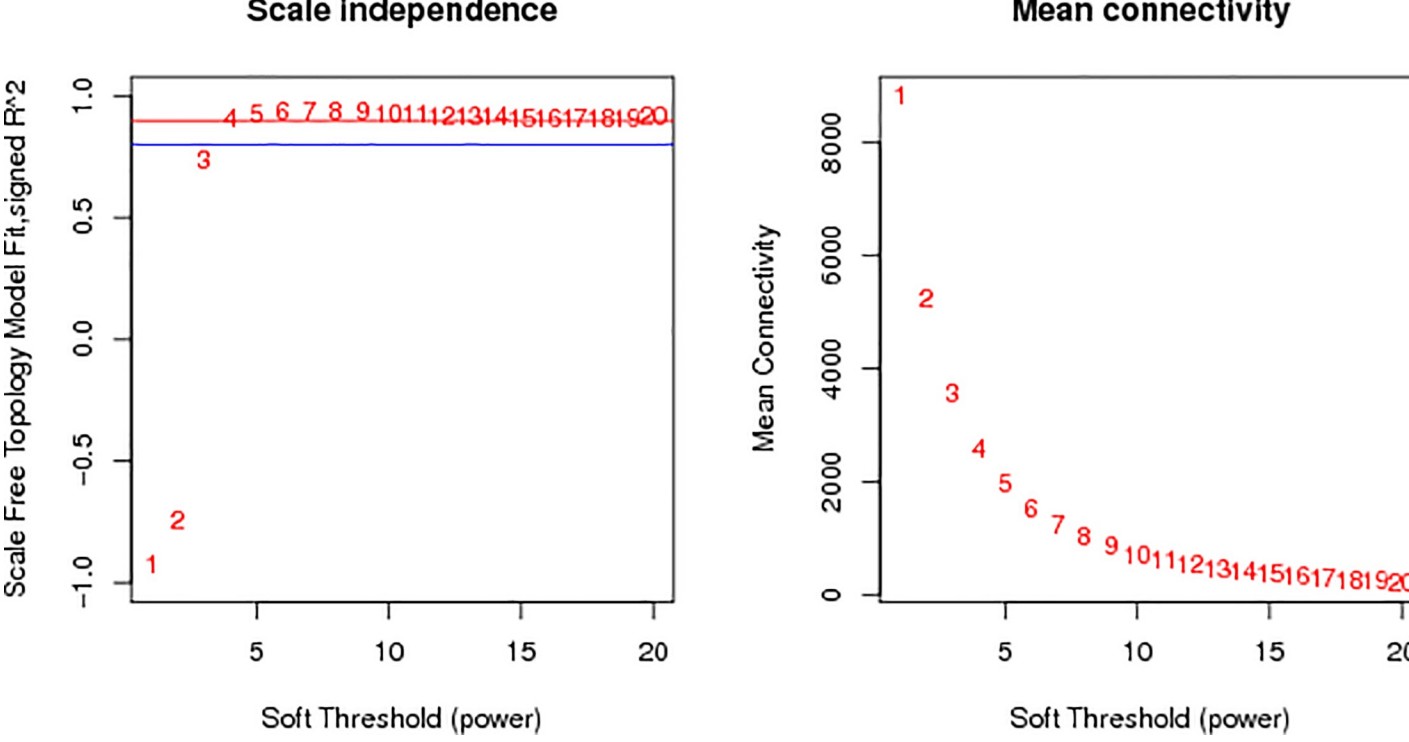

**Fig 2. Power value curve.**

### Screening key candidate genes in specific target modules

The correlation coefficients between genes and the eigenvalues of each module were calculated using the cor function (Pearson's correlation coefficient), and the sum of the correlation coefficients between each gene and other genes was calculated in the adjacency matrix to obtain the connectivity of each gene. Previous studies showed that genes with high connectivity and genes related to MEs were potential candidate hub genes, and that they represented the overall level of the module [11, 12]. To identify the candidate hub gene of the blue module, we created a Wayne map based on the intersections of genes with connectivity ≥900 in the blue module and genes with correlation ≥0.9 with MEblue as candidate hub genes in the blue module (Fig 5).

### Enrichment analysis of GO/KEGG functions for key candidate genes in specific target modules

We further analyzed the 584 candidate hub genes in the blue module by GO/KEGG enrichment analysis, with a focus on enrichment of GO Biological Process and KEGG signal pathways, which could directly indicate if the enriched genes were related to the occurrence and development of skin follicles. Candidate hub genes were significantly enriched in 11 GO terms (Fig 6). Skin development, hair cycle, and tissue development were directly involved in the development of skin hair follicles. In the GO cellular component, 22 GO terms were enriched in the candidate genes (Fig 7), with cell-cell junction being significantly enriched. This suggested that the genes enriched in this GO term might regulate the degree of cell-cell compactness; however, this function is limited to cell-cell connections, and does not indicate cell-cell communication and material exchange [13, 14]. The candidate genes were enriched in 24 GO molecular function terms (Fig 8). Negative regulation of catalytic activity was significantly

## Cluster Dendrogram

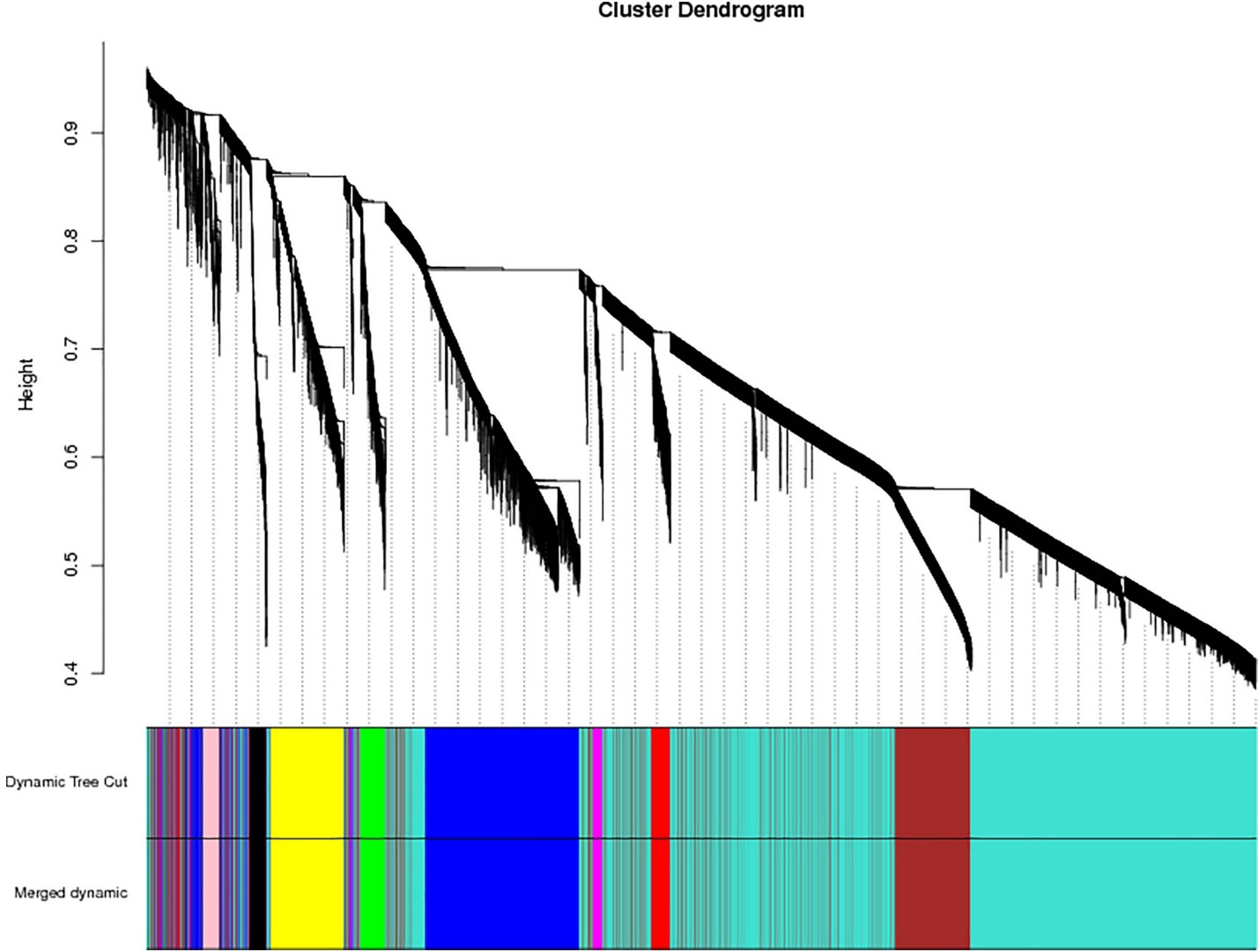

**Fig 3. Module hierarchical clustering tree.** Dynamic tree cut indicates module divided according to clustering results; merged dynamic indicates module divided according to similarity of the module. Analysis was carried out according to the merged module. Vertical distance in tree diagrams represents distance between two nodes (between genes); horizontal distance is meaningless.

enriched, and we speculated that genes enriched in this GO term might be involved in inhibiting the expression of genes related to hair follicle development; e.g. RALT/MIG6 inhibits the abnormal expression of epidermal growth factor receptor, thus preventing epithelial cells from transforming into cancer cells [15]. Candidate hub genes were enriched in 47 signaling pathways in KEGG analysis (Fig 9), including three significantly enriched pathways: fatty acid degradation, estrogen signaling pathway, and Staphylococcus aureus. These three signaling pathways showed no relevant report on the occurrence and development of skin hair follicles.

### Screening of key hub genes in mature Inner Mongolian cashmere goat fetal skin follicles

GO Biological Process enrichment identified 89 genes that were enriched in skin development, the hair cycle, and tissue development, including *WNT10A*, *DNASE1L2*, *KRT25*, *HOXC13*,

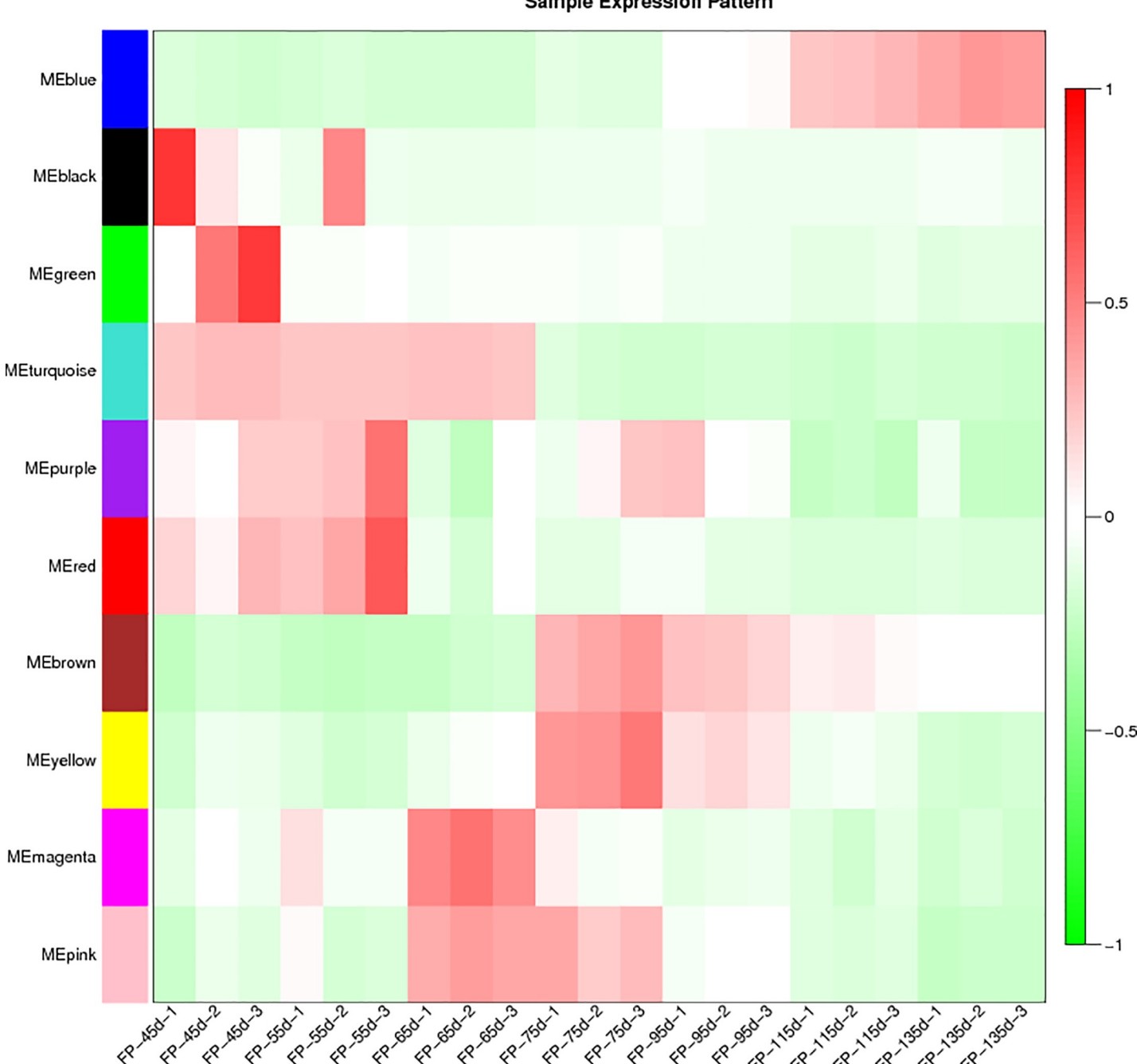

**Fig 4. Heat map of sample expression pattern.** Abscissa represents the sample and ordinate represents the module, based on module eigenvalues. Red: High expression; green: Low expression.

*KRT27*, *FOXN1*, *TMEM79*, *LOC102177231*, and *LOC102176999*, which were enriched in all three GO terms (Fig 10). Notably, candidate hub genes were enriched in 12 KEGG signaling pathways directly involved in skin hair follicle development (Fig 9). Although there were no significant differences in the enrichment of KEGG signaling pathways, they are of great biological significance in the occurrence and development of skin hair follicles and hair morphogenesis. The 12 signaling pathways were enriched in 56 genes, including *AREG*, *FZD10*, *EFNA3*,

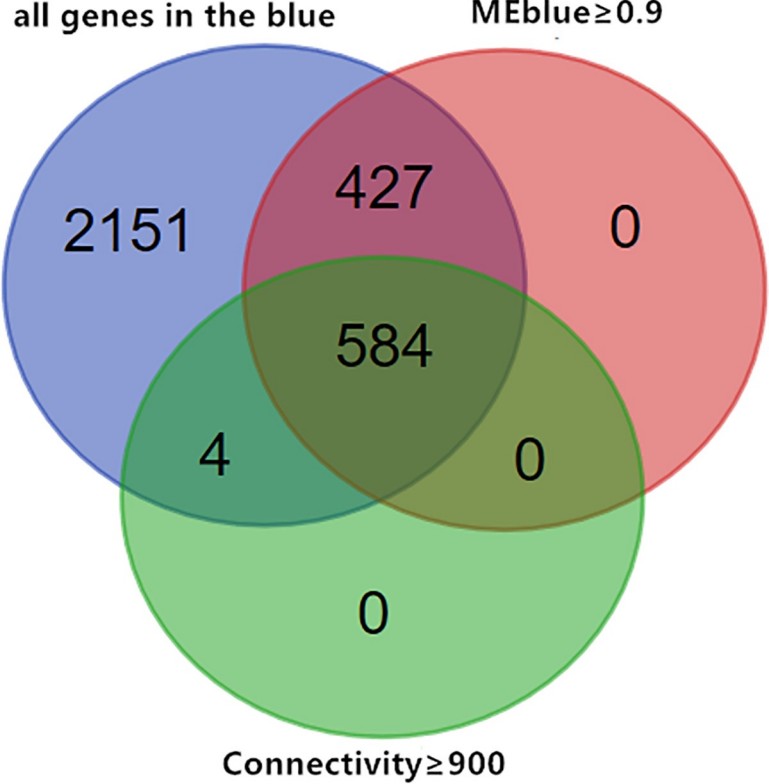

**Fig 5. Venn diagram of key candidate genes for specific target modules.**

*EPHA2*, *BMP4*, *FZD7*, *WNT10A*, *LOC102175216*, and *PLA2G2F*, which were enriched in more than three signaling pathways (Fig 11). *WNT10A* was the only gene enriched in all three GO terms and all three KEGG signaling pathways (Fig 12). We therefore identified *WNT10A* is the key hub gene in the development and maturation of skin hair follicles in fetal Inner Mongolian cashmere goats.

## Quantitative identification of core and key genes by real-time fluorescence

We selected 13 genes for QPCR verification, among which the expression trends of the following genes were strongly correlated with the RNA-Seq results (Fig 13): *ACTN4* (Fig 13A), *APRC2* (Fig 13B), *EFNA3* (Fig 13D), *FOXN1* (Fig 13E), *FZD7* (Fig 13F), *FZD10* (Fig 13G),

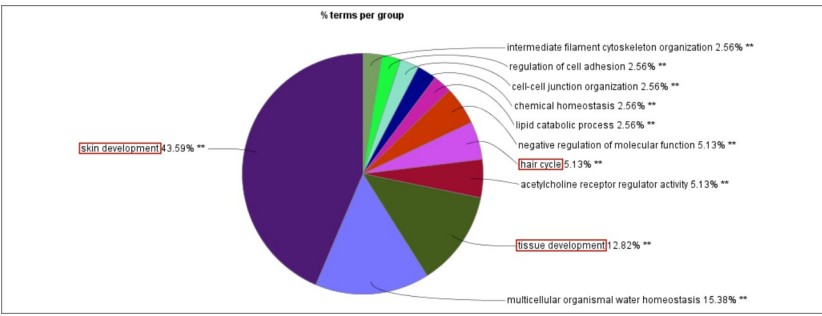

**Fig 6. GO-biological process.** \*\*P<0.01. Red box: GO terms directly involved in the development of skin hair follicles.

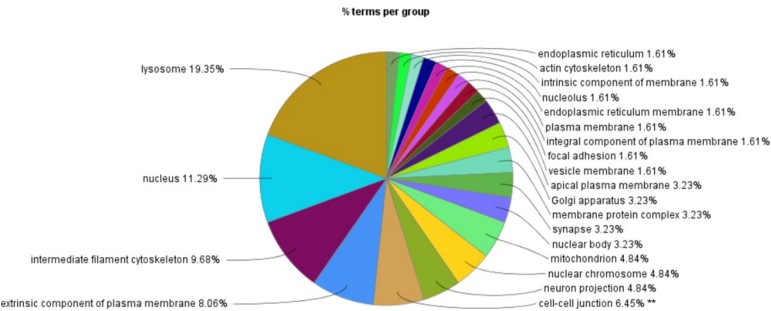

**Fig 7. GO-cellular component.** \*\*P<0.01.

*LAMC3* (Fig 13H), *LOC102180576* (Fig 13I), *TRPV3* (Fig 13L), and *WNT10A* (Fig 13M). The expression trends for *MAP2K3* (Fig 13J) and *NSUN2* (Fig 13K) showed strong positive correlations with RNA-Seq, and the expression trend for *EFNA1* (Fig 13C) was moderately positively correlated with RNA-Seq.

## Discussion and summary

In this study, we used previous RNA-Seq results for different fetal stages of Inner Mongolian cashmere goats to determine 10 co-expression network modules using WGCNA, and successfully mined the specific module related to mature skin and hair follicle development based on the ME. WGCNA divides modules into soft thresholds, which reflect the effectiveness of biological networks more effectively than hard thresholds [11]. Finally, GO/KEGG functional enrichment analysis of candidate key genes identified *WNT10A* as the key core gene for fetal skin follicle development and maturation in Inner Mongolian cashmere goats.

The two most important concepts used to describe nodes in the topological overlap matrix constructed by WGCNA are the correlations between genes and MEs and gene connectivity. The characteristic value of a module represents the overall level of the module, such that a closer correlation between the gene expression level and the ME indicates that the gene may be the potential hub gene for the module. Gene connectivity is the sum of the correlation coefficients between the gene and other genes, and higher gene connectivity indicates more associated genes, further suggesting that the gene of interest may be a hub gene in the module [10–12].

Although the fatty acid degradation, estrogen signaling pathway, and *Staphylococcus aureus* infection KEGG pathways were significantly enriched among the 584 candidate hub genes, there is no relevant report on the occurrence and development of skin hair follicles. Fatty acids

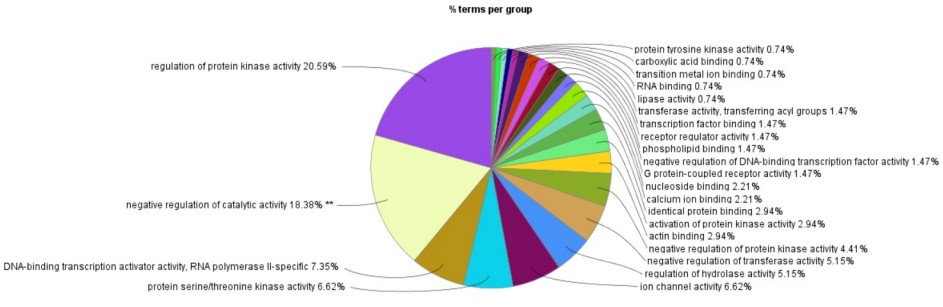

**Fig 8. GO-molecular function.** \*\*P<0.01.

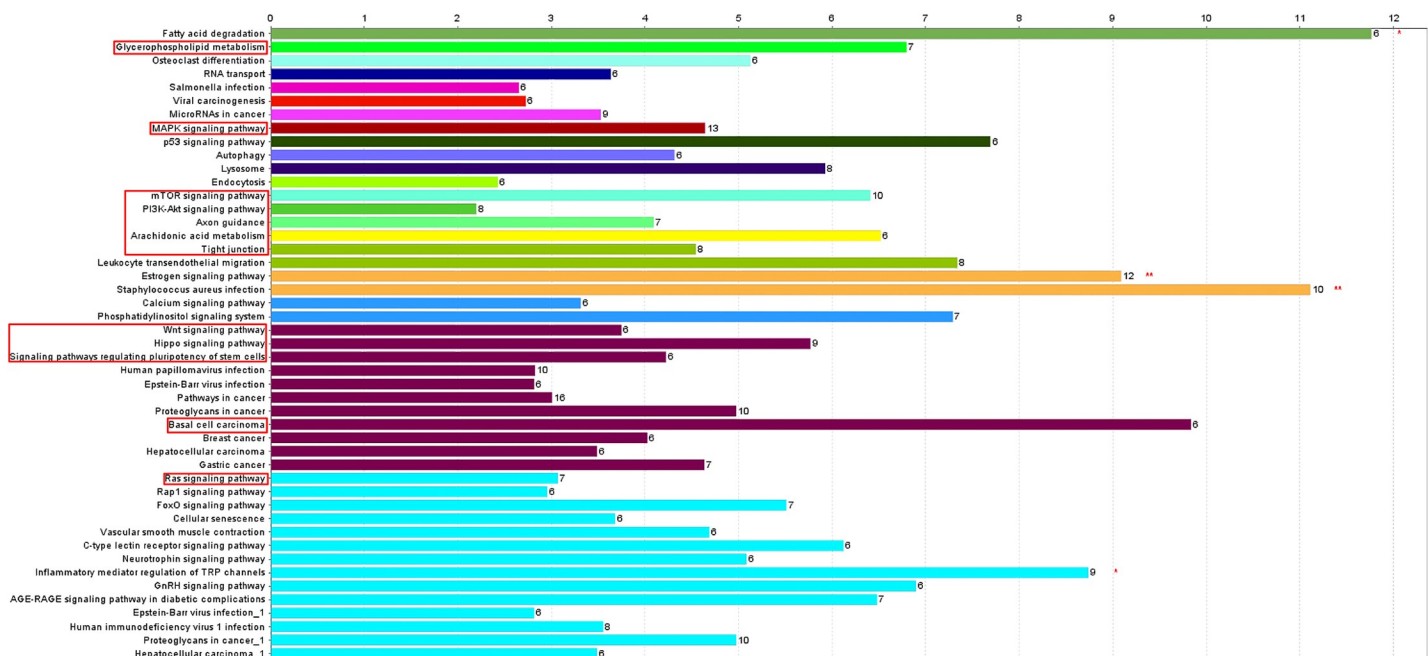

**Fig 9. KEGG signal pathway enrichment diagram.** Abscissa indicates percentage of genes enriched by each signaling pathway relative to total genes of each signaling pathway; numbers to right of columns indicate number of genes enriched by each signaling pathway; red boxes: Signaling pathways involved in development of skin hair follicles. *P<0.05, **P<0.01.

on the skin surface are mainly composed of sebum and a small amount of intercellular lipids. They are largely responsible for maintaining the normal skin barrier function, and the degradation of fatty acids by the skin can maintain normal skin metabolism [16]. Studies of wound healing showed that dynamic changes in estrogen levels in line with aging and the menstrual cycle affected wound healing, because estrogen combines with receptors on the skin to activate the estrogen signaling pathway and accelerate wound healing [17, 18]. Epithelial cells contain the skeletal protein keratin, which maintains their basic morphology. At the same time, many pathogenic microorganisms, including *Staphylococcus aureus*, enter the body via the skin [19, 20]. We therefore focused on 12 signaling pathways with biological significance in the development of skin hair follicles.

During glycerophospholipid metabolism, phospholipase degrades and oxidizes bioactive phospholipids in the epidermis, enabling epithelial cells to perform their normal cellular functions and maintain the epidermal structure [21]. However, disruption of glycerophospholipid metabolism results in oxidative damage to the epidermis, such as psoriasis [22]. Fibroblast growth factor (FGF) 21 knockout mice showed slower hair regeneration and had significantly fewer hair stems than wild-type mice, possibly related to the role of FGF21 as the key gene in the MAPK and phosphoinositide 3-kinase/Akt signaling pathways [23]. The importance of these signaling pathways in hair follicle development was also shown in a study of hair morphology of Hu wool, The Ras signaling pathway may also be involved in hair follicle development [24]. Hair follicles progress from the resting stage to the growth stage as a result of the rapid differentiation of hair follicle stem cells, related to activation of the mTOR signaling pathway [25]. The formation of skin hair follicles is related to the fine rearrangement of various cells, notably including the appearance of dermal aggregations induced by epithelial cells. Genes involved in axon guidance are highly expressed in epithelial cells and the dermis, suggesting that axon guidance might play a key role in this process [26]. Arachidonic acid

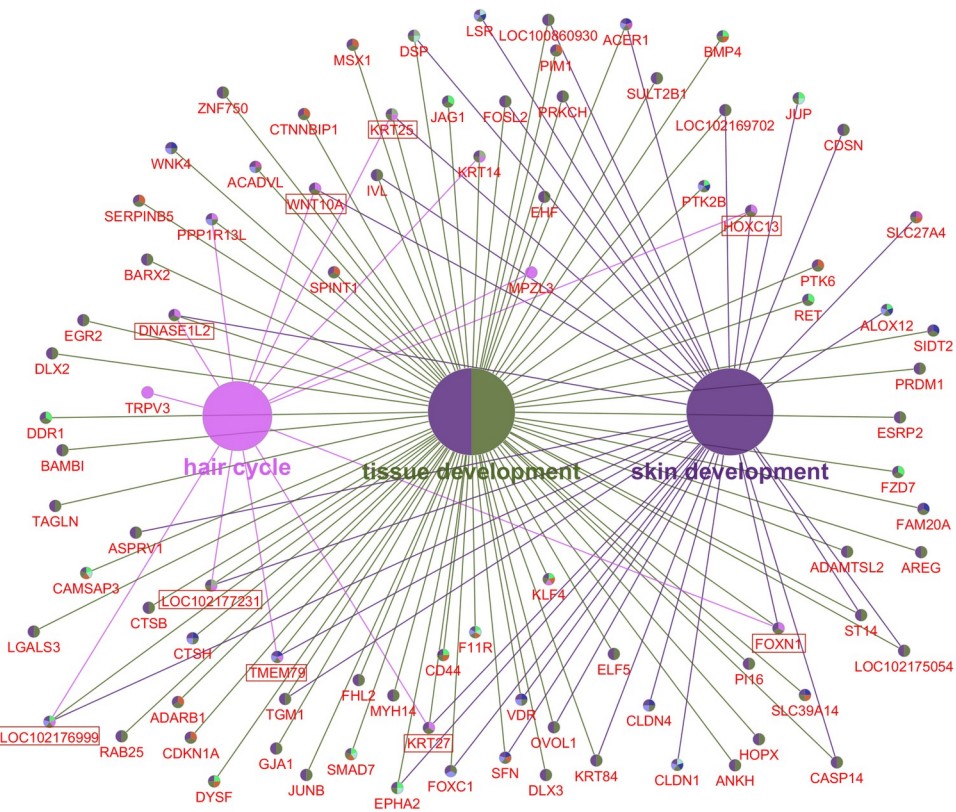

**Fig 10. Gene enrichment map of skin follicle development and GO-biological process.** Large nodes: GO terms directly involved in the biological process of skin hair follicle development; small nodes: Enriched genes; red boxes: Enriched to three GO terms.

metabolism can produce prostaglandins $D_2$ (PGD2), $E_2$ (PGE2), $F_{2-\alpha}$ (PGf2α) and $I_2$, and thromboxane A2 through different catalytic reactions. PGE2 protects irradiated mice from hair loss, PGF2α promotes the growth of human eyelashes, while PGD2 inhibits hair growth [27]. Tight junctions are an important prerequisite for tight intercellular connections, as well as providing a means of intercellular communication, and also playing an important role in different stages of the hair cycle and the process of hair follicle development [28, 29]. The Hippo signaling pathway is a highly conserved signaling pathway, which is targeted by the miR-200 family of microRNAs to regulate the cell fate (cell migration and adhesion) of hair embryo, resulting in normal hair development [30]. Pluripotent stem cells have self-renewal ability, and 3D culture of pluripotent stem cells *in vitro* produced skin epidermis and dermis, together with new hair follicles [31]. The Hedgehog and Wnt signaling pathways are important components of basal cell carcinoma and their interaction has been shown to lead to abnormal differentiation of basal cells and eventually to basal cell carcinoma [32]. The Wnt signaling pathway is an important part of many signaling pathways related to hair follicle development, such as Hippo signaling, basal cell carcinoma, and signaling pathways regulating the pluripotency of stem cells. In addition, Wnt signaling was shown to play an important role in hair follicle differentiation and maturation in the development of wool follicles in Northern Shaanxi cashmere goats [33]. These studies thus demonstrated the importance of the Wnt signaling pathway in hair follicle development.

Mammalian skin and its appendages are derived from mesoderm and ectoderm during embryogenesis [34, 35]. The embryo surface comprises a monolayer of epithelial cells, while

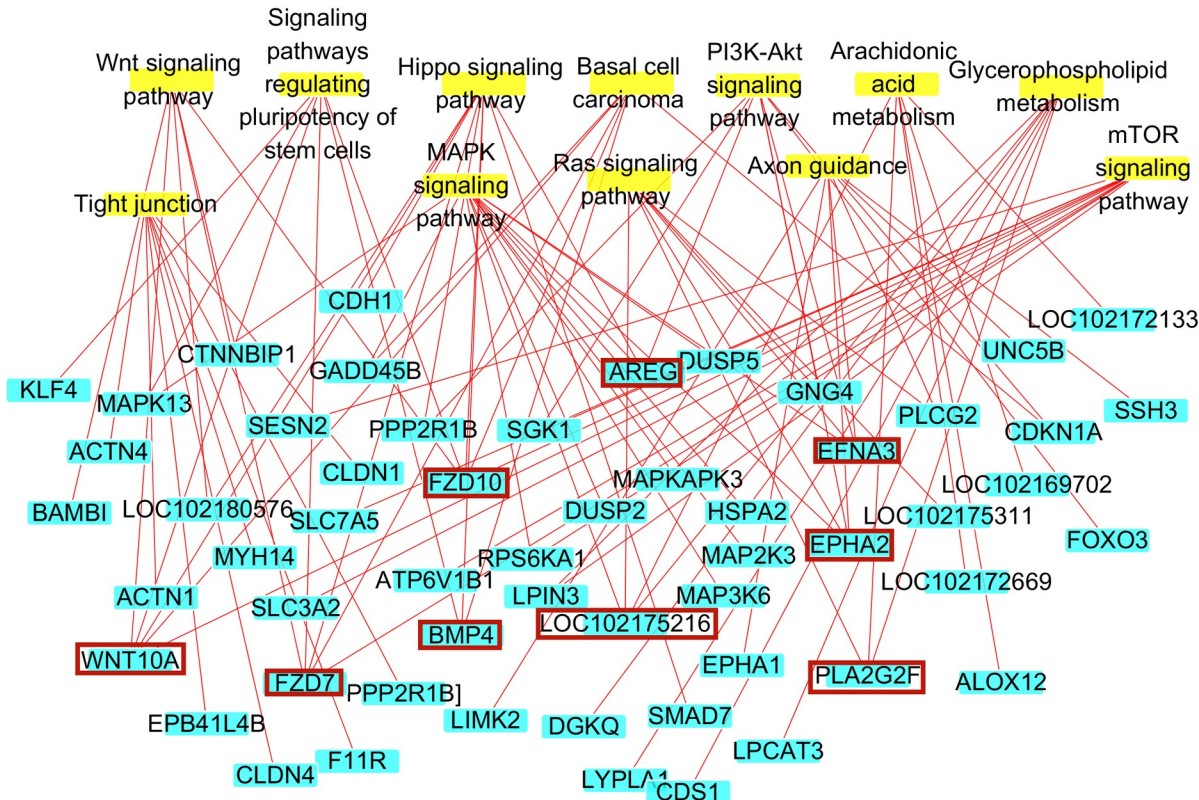

**Fig 11. Gene enrichment diagram of KEGG signaling pathway in skin hair follicle development.** Nodes with yellow background: Signal pathways; nodes with blue background: Genes; red boxes: Genes enriched in at least three signal pathways.

the dermis is composed of fibroblasts, and hair follicles are induced by the interaction of the epidermis and dermis [35, 36]. The exchange of information between the epidermis and the dermis during hair follicle development involves a family of secreted signaling molecules, of which the Wnt family of proteins may be the earliest and most critical in epidermal development [35, 37–39]. Wnt family members (19 members in humans and mice) and their receptor, Frizzled (FZD) protein (10 members in humans and mice), form a large family, and the network they construct can control several different cell pathways, including their proliferation, fate, shape, and movement [40, 41].

Some members of the Wnt family have different expression patterns during skin development, and several candidates are related to the morphogenesis of hair follicles, such as *WNT3*, *WNT10A*, and *WNT10B* [42–44]. *WNT10A* is expressed in dermal papilla cells and in the adjacent follicular epithelium during hair growth [44] and can regulate the cell fate pattern [45]. Although several *WNT* family genes have been found in epithelial cells on the surface of the embryo, only the transcripts of *WNT10A* and *WNT10B* were located specifically in the basal layer of the hair follicle and the inner root sheath precursor in the matrix, while *WNT10A* was even expressed in the more-differentiated inner root sheath cells. The function of *WNT10A* is thus considered as part of the "first epithelial signal" in hair follicle morphogenesis [44]. In addition to its role in hair follicle development, *WNT10A* is also involved in acne in humans, and its dislocated alleles have a protective effect on acne [46].

The Wnt/β-catenin signaling pathway is generally divided into classical and non-classical pathways. In the classical pathway, Wnt ligands bind to low-density lipoprotein receptors,

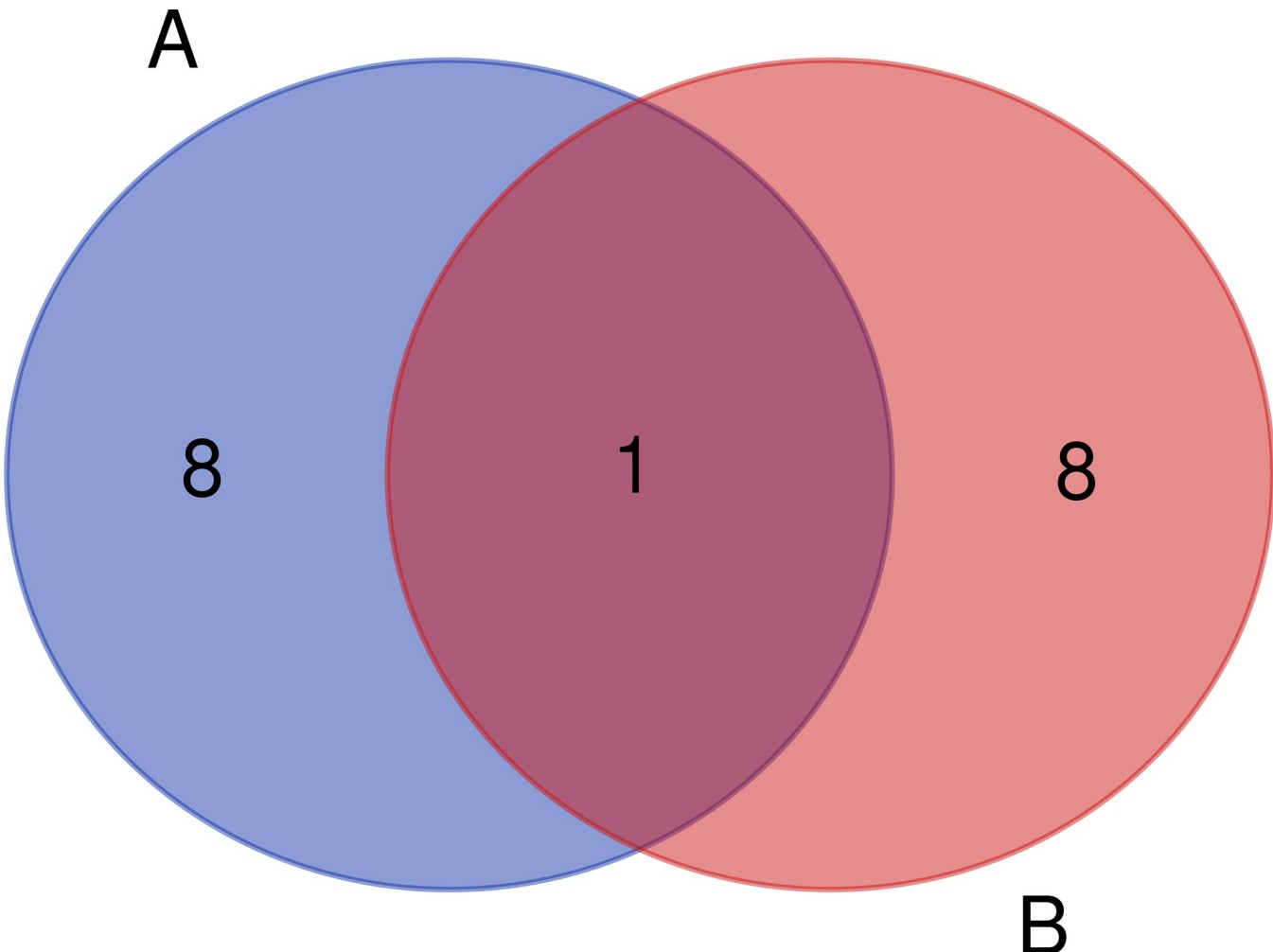

**Fig 12. Venn diagram of GO/KEGG enrichment analysis.** A: The genes marked by the red box are all enriched to 3 GO terms. B: The gene enriched to $\geq 3$ signal pathways is marked by the red frame.

associated protein LRP-5 and LRP-6 co-receptors, and FZD receptors, leading to the activation of disheveled (DSH) and inhibition of the β-catenin degradation complex to stabilize the cytoplasm. β-catenin then accumulates in the cytoplasm, enters the nucleus, and activates Wnt-regulated targets through interaction with T cell factor (TCF) family transcription factors, with subsequent recruitment of co-activator genes [40]. In the non-classical Wnt/β-catenin signaling, knypek (a member of the glypican family of heparan sulfate proteoglycans) enhances the binding of Wnt to the FZD receptor [47], without involvement of LRP5 and LRP6. DSH is activated by non-classical Wnt signaling via a different protein domain from the classical Wnt/β-catenin signaling domain [48]. Non-classical Wnt signaling does not need β-catenin or Lymphoid Enhancer Factor/T-Cell Factor, and in contrast, DSH is activated by the bridging molecule Daam1 [49]. Activated DSH can cause calcium flux and activation of calcium-sensitive kinase, protein kinase C, and calcium/calmodulin-dependent protein kinase II [50, 51]. However, it is unclear if Rho1 activation and calcium flux activation occur in the same pathway or in different branches of the non-classical FZD-DSH pathway [52]. In some cases, non-classical Wnt signaling can be blocked by the Wnt/β-catenin pathway.

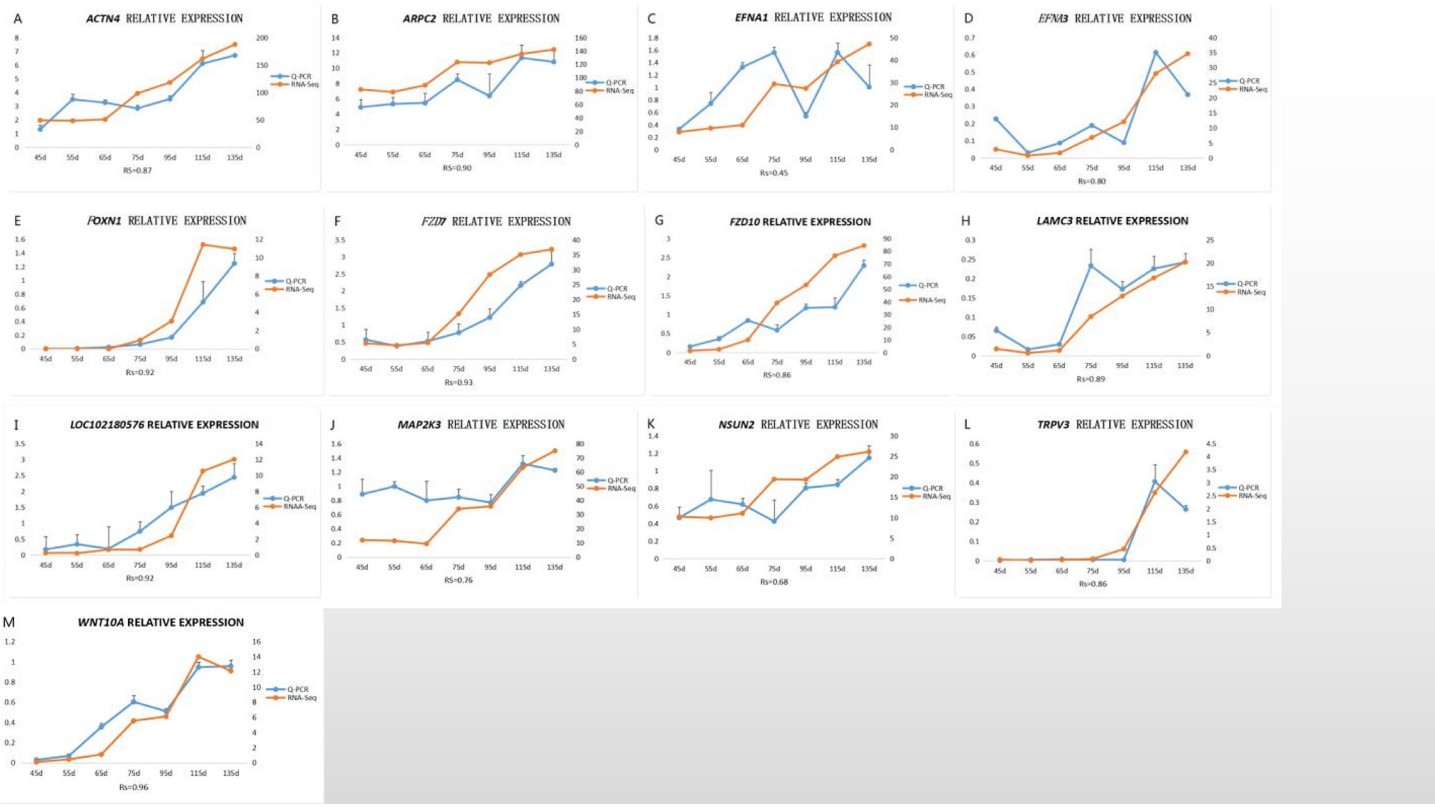

**Fig 13. Fluorescent QPCR to verify the sequencing results.** Spearman's rank correlation: 0.8 < |Rs| < 1 extremely strong correlation, 0.6 < |Rs| < 0.8 strong correlation, 0.4 < |Rs| < 0.6 medium correlation, 0.2 < |Rs| < 0.4 weak correlation, 0 < |Rs| < 0.2 no correlation.

Wnt-10A not only plays a role in the development of skin hair follicles, but also has an important role in the development of other ectodermal structures. The ectoderm is the outer-most layer of the embryo, which develops into the nervous system, sensory system, and the skin and skin appendages. Ectodermal dysplasia is a congenital disease that affects the mor-phogenesis of the skin, hair, nails, teeth, sweat and sebaceous glands, submucosa, and breast. *WNT10A* is an important gene in ectodermal development, and its mutation may result in ectodermal dysplasia [41], with the main symptoms of abnormal development of dentin and nails and the appearance of cysts and erythema along the edge of the eyelid [53, 54]. During tooth development, the specific expression of *WNT10A* in epithelial cells can induce tooth-root bifurcation, and it is therefore also considered as a key gene in the process of tooth differ-entiation, *WNT10A* also controls the proliferation of adjacent mesenchymal cells by regulating the expression of Wnt4 in mice [55, 56].

*WNT10A* also plays an important role in cancer. Its absence can delay the healing of dorsal skin wounds in mice [57], while tumors are considered as "unhealable" wounds. Related to the mechanism of *WNT10A* in skin wound healing, its absence can inhibit the expression of colla-gen, thereby preventing tumor occurrence [58]. Colorectal cancer is one of the main causes of cancer-related deaths worldwide, and *WNT10A* may be involved in the occurrence of colorec-tal cancer by activating Wnt/β-catenin signaling [59]. Notably, hypermethylation of *WNT10A* can also be used as a diagnostic marker for colorectal cancer [60].

Further studies are needed to verify the functions of the candidate core gene *WNT10A* using methods such as QPCR, western blot, and transgenic methods, in order to gain a deeper

understanding of the relationship between the genes and modules identified in this study and the mature stage of skin hair follicles in fetal Inner Mongolian cashmere goats. Revealing the key genes in mature skin hair follicles will help to elucidate the regulatory network involved in skin hair follicle maturation, with beneficial implications for the breeding and yield of Inner Mongolian cashmere goats.

## Supporting information

**S1 Table. Transcriptome data.** The gene expression data for each sample was normalized using the fragments per kilobase of exon per million fragments method.
(XLS)

**S2 Table. Connectivity of all genes.**
(XLSX)

**S3 Table. Module eigenvalue.**
(XLSX)

**S4 Table. Correlation between gene and module eigenvalue.**
(XLSX)

**S1 File.**
(ZIP)

## Author Contributions

**Conceptualization:** Zhihong Wu, Erhan Hai, Yanjun Zhang, Jinquan Li.

**Data curation:** Zhihong Wu, Erhan Hai, Zhengyang Di, Yanjun Zhang.

**Formal analysis:** Zhihong Wu.

**Funding acquisition:** Yanjun Zhang, Jinquan Li.

**Investigation:** Zhihong Wu, Erhan Hai.

**Methodology:** Zhihong Wu, Rong Ma.

**Software:** Zhihong Wu, Erhan Hai, Fangzheng Shang.

**Supervision:** Zhihong Wu.

**Validation:** Zhihong Wu.

**Visualization:** Zhihong Wu, Erhan Hai.

**Writing – original draft:** Zhihong Wu.

**Writing – review & editing:** Erhan Hai, Yu Wang, Min Wang, Lili Liang, Youjun Rong, Jianfeng Pan, Wenbin Wu, Rui Su, Zhiying Wang, Ruijun Wang, Yanjun Zhang, Jinquan Li.

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
