## [Decision Letter · Decision Letter 0]

3 Sep 2020

PONE-D-20-17381

Using WGCNA（weighted gene co-expression network analysis） to excavate the hub genes of skin hair follicle development in fetus stage of Inner Mongolia Cashmere goat

PLOS ONE

Dear Dr. Zhang,

Thank you for submitting your manuscript to PLOS ONE. After careful consideration, we feel that it has merit but does not fully meet PLOS ONE’s publication criteria as it currently stands. Therefore, we invite you to submit a revised version of the manuscript that addresses the points raised during the review process.

We look forward to receiving your revised manuscript.

Kind regards,

Naresh Doni Jayavelu, Ph.D

Academic Editor

PLOS ONE

Journal Requirements:

Additional Editor Comments (if provided):

1. Address reviewers all comments.

2. Provide the high resolution figures.

3. Language should be proof-read by the native English speaker.

Reviewers' comments:

Reviewer's Responses to Questions

**Comments to the Author**

1. Is the manuscript technically sound, and do the data support the conclusions?

Reviewer #1: Partly

Reviewer #2: Yes

2. Has the statistical analysis been performed appropriately and rigorously? 

Reviewer #1: I Don't Know

Reviewer #2: I Don't Know

3. Have the authors made all data underlying the findings in their manuscript fully available?

Reviewer #1: No

Reviewer #2: Yes

4. Is the manuscript presented in an intelligible fashion and written in standard English?

Reviewer #1: No

Reviewer #2: No

5. Review Comments to the Author

Reviewer #1: Summary of the research

In this manuscript from Whu et al. Authors generated temporal transcriptome for Inner mongolican cashmere goat fetal skin. Transcriptomes were generated specifically for 7 time points during the fetal growth period (45, 55, 65, 75, 95, 115 and 135 days). Authors further employed the WGCNA method to understand the hub genes for hair follicle development in the fetal skin transcriptome. They also used the GO and KEGG analysis to understand the modules for their functionality.

Major comments

Authors should revise the language to improve the readability and thoroughness of the manuscript. For example authors discuss WGCNA in the title, abstract and introduction but why these transcriptomic study is important (for example with respect to improving the livestock or to improve basic understanding of the hair follicle development in cashmere goat). Author needs to mention what is the current status of research in the field, what are the limitations or gaps in the field and how finding from the current manuscript can help fill those gaps in the understanding.

Rational to select blue module based on module wise gene expression on 115 and 135 days is not very clearly explained. As the development process is a continuum, gene expression patterns at early stages may govern quality of hair as much as the later stages. It would be important to give clear supporting reasons to select the blue modules

GO biological process enrichment: current study looks into transcriptome of fetal skin thus having skin development gene enrichment , hair cycle enrichment is convincing but does that stand true for only blue module (selected for further analysis) or for many other modules and highly expressed genes. In other words these GO enrichment is specific to blue module or other module (highly expressed genes) as well

Rational behind GO cellular component and Molecular function is not very clear from the discussion or results sections. This can be discussed further why regulation of protein kinase is important or why lysosomal cellular components are significantly enriched cellular components.

Figure 11 in KEGG signaling pathway enrichment Authors did not discuss/comment about statistically significant pathways like estrogen signalling, staphylococcus aureus infection, does this indicate pre existing infection from the collected sample or how this variable affects the normal growth of skin follicles. Also cherry picking of pathway which are very general (may be involved in skin development but not exclusive to skin development) like axon guidance, mTOR signaling pathway, Wnt signaling, Hippo signalling these are very important pathways for in general development. They may be involved in skin development as well but cherry picking them over statistically significant pathways from KEGG does not seem to be very logical. Authors do give reference but it would be clear to also discuss those references ( ref 11-24) with respect to their finding how they suggest that pathway is involved and important in hair follicle development.

Selection of grouping for figure 13 is also not very clear and needs further explanation. Thus major finding about wnt10a does not seem to evolve naturally. Authors forgot to mention similar study which indicates important role of wnt10a in hair follicle development from Gao et al (https://doi.org/10.1371/journal.pone.0151118)

Minor comments

Diagramatic representation of primary and secondary hair follicle and how it is related to cashmere wool vs cashmere production can be helpful

For example authors mention “development rule of inner mongolia cashmere skin follicle” can be represented as a graphical representation and need to discuss it in introduction so later on rational towards it can be easily understood

Figure legends should be clear and detailed, also figure axis should be readable for example figure 2 legends are not very clears, Figure 5 , Figure 7 , Figure 11 axis are not clearly readable

It would be more transparent if authors can provide their R scripts used for the analysis for quick check

Reviewer #2: Comments

Minor issues

1) Authors say that data is available without restrictions but I did not see a link to the data

2) The English and grammar is not upto the scientific standards. Needs a lot of correction.

3) Figure legends are not in a standard format

Major issues

1) Authors do not describe the sequencing data analysis. Which tools were used in processing and normalization etc. No description of any statistics that was performed and should have been performed the assess the gene expression levels of the gene of interest from the sequencing data. The raw data should be made available so people can reproduce the results.

6. PLOS authors have the option to publish the peer review history of their article (what does this mean?). If published, this will include your full peer review and any attached files.

Reviewer #1: No

Reviewer #2: No

---

## [Author Response · Author response to Decision Letter 0]

20 Oct 2020

Dear editors and reviewers:

Thank you for your letter and the reviewers’ comments on our manuscript entitled "Using WGCNA（weighted gene co-expression network analysis) to identify the hub genes of skin hair follicle development in fetus stage of Inner Mongolia Cashmere goat" (ID：PONE-D-20-17381). Those comments are very helpful for revising and improving our paper, as well as the important guiding significance to other research. We have studied the comments carefully and made corrections which we hope meet with approval. The main corrections are in the manuscript and the responds to the reviewers’ comments are as follows (the replies are highlighted in blue ).In addition, we have added several important supporting information

Replies to the reviewers’ comments:

Reviewer #1:

1. Authors should revise the language to improve the readability and thoroughness of the manuscript. For example authors discuss WGCNA in the title, abstract and introduction but why these transcriptomic study is important (for example with respect to improving the livestock or to improve basic understanding of the hair follicle development in cashmere goat). Author needs to mention what is the current status of research in the field, what are the limitations or gaps in the field and how finding from the current manuscript can help fill those gaps in the understanding.

Response：We have re-written Introduction according to the Reviewer’s suggestion.We are sorry that our English writing level has troubled the reviewers. We have submitted our manuscripts to international science editing(http://www.internationalscienceediting.com) for polishing as suggested by the reviewers.

2. Rational to select blue module based on module wise gene expression on 115 and 135 days is not very clearly explained. As the development process is a continuum, gene expression patterns at early stages may govern quality of hair as much as the later stages. It would be important to give clear supporting reasons to select the blue modules.

Response：According to your first suggestion, we re wrote the introduction, and based on this, we extended the reason why we chose blue module as the research object.Most research on skin and hair follicles in cashmere goats has focused on changes in the primary and secondary hair follicles during their growth, degeneration, and resting stages, and on the mechanisms of their related genes. The fetal development of primary and secondary hair follicles has been less well studied, and studies of fetal skin and hair follicle development have largely considered the occurrence of primary and secondary hair follicles (fetal period 45–65 days). However, there have been no reports on the maintenance of hair follicle structure and skin hair follicle homeostasis by epithelial cells and fibroblasts during the development and maturation of the primary and secondary hair follicles (115–135 days), even though the stability of the epithelial cells and fibroblasts has been shown to be a key requirement for the hair follicle to enter the hair cycle and to maintain the normal structure of the hair follicle in other mammals

3. GO biological process enrichment: current study looks into transcriptome of fetal skin thus having skin development gene enrichment , hair cycle enrichment is convincing but does that stand true for only blue module (selected for further analysis) or for many other modules and highly expressed genes. In other words these GO enrichment is specific to blue module or other module (highly expressed genes) as well.

Response：I'm glad the reviewers noticed this.At the beginning, we carried out go / KEGG enrichment analysis for all modules as recommended by the review. However, we found that there was a big disadvantage in this way. The selected gene was likely not a hub gene. This method was similar to the conventional transcriptome data analysis, which also lost the significance of WGCNA method. WGCNA is based on the scale-free network. Some studies have confirmed that the metabolic network and protein interaction of organisms conform to the scale-free network. In scale-free networks, the connectivity P (k) between genes follows the power law distribution, P (k) ~ k-y. A key feature of scale-free networks in organisms is that there are some nodes with high connectivity, which participate in a large number of metabolic reactions. Through a large number of connections, these nodes form a single integrated network. Compared with random networks, scale-free networks are more stable. Because of the characteristics of scale-free network, we can find the core target in drug therapy to improve the efficacy of drugs, or find the related central genes in some biological phenomena. And these central genes can also represent the overall level of their modules. Generally, gene connectivity and the correlation between gene and module eigenvalues are important indicators for screening central genes. Therefore, we first calculated the connectivity of all genes in the blue module and the eigenvalues of genes and the blue module to obtain candidate hub genes in the blue module, and further through the go / KEGG enrichment analysis, we obtained the hub genes related to hair follicle development.）

4.Rational behind GO cellular component and Molecular function is not very clear from the discussion or results sections. This can be discussed further why regulation of protein kinase is important or why lysosomal cellular components are significantly enriched cellular components.

Response：We have made correction according to the Reviewer’s comments.（3.4 Enrichment analysis of GO/KEGG functions for key candidate genes in specific target modules）

5.Figure 11 in KEGG signaling pathway enrichment Authors did not discuss/comment about statistically significant pathways like estrogen signalling, staphylococcus aureus infection, does this indicate pre existing infection from the collected sample or how this variable affects the normal growth of skin follicles. Also cherry picking of pathway which are very general (may be involved in skin development but not exclusive to skin development) like axon guidance, mTOR signaling pathway, Wnt signaling, Hippo signalling these are very important pathways for in general development. They may be involved in skin development as well but cherry picking them over statistically significant pathways from KEGG does not seem to be very logical. Authors do give reference but it would be clear to also discuss those references ( ref 11-24) with respect to their finding how they suggest that pathway is involved and important in hair follicle development.

Response：We are very sorry for our neglected to discuss the role of these signaling pathways in hair follicle development, and We have re-written this part in discussion and summary according to the Reviewer’s suggestion (by reviewing a large number of literatures, we reduced the number of previously identified 13 signal pathways related to hair follicle development to 12, but this did not affect the final results.)

Estrogen signaling pathway and Staphylococcus aureus infection were significantly enriched, which did not indicate that our samples were infected. Firstly, estrogen accelerators cubaneous soil healing by promoting promotion promotion of epithelial keratinocytes via ERK / Akt signaling pathway, but there is not enough evidence to show that it is related to hair follicle development. We found that almost all the genes enriched in Staphylococcus aureus infection are keratin family. Keratin family is the skeleton protein of epithelial cells. They maintain the normal morphology and adhesion of epithelial cells. The main way of Staphylococcus aureus infection is that its’ surface proteins can bind with proteins related to cell adhesion, and the most important ligand is keratin family But that doesn't mean our samples are infected.

6.Selection of grouping for figure 13 is also not very clear and needs further explanation. Thus major finding about wnt10a does not seem to evolve naturally. Authors forgot to mention similar study which indicates important role of wnt10a in hair follicle development from Gao et al (https://doi.org/10.1371/journal.pone.0151118).

Response：We have made correction according to the Reviewer’s comments.(3.5 Screening of key hub genes in mature Inner Mongolian cashmere goat fetal skin follicles).And we are very grateful to the reviewers for providing us with the relevant literature, which has increased the persuasive power of our research results.

7.Diagramatic representation of primary and secondary hair follicle and how it is related to cashmere wool vs cashmere production can be helpful. For example authors mention “development rule of inner mongolia cashmere skin follicle” can be represented as a graphical representation and need to discuss it in introduction so later on rational towards it can be easily understood.

Response：According to the suggestion of the reviewers, we have drawn a map of the development of the skin and hair follicles of Inner Mongolia cashmere goats. This article has been greatly enhanced

8.Figure legends should be clear and detailed, also figure axis should be readable for example figure 2 legends are not very clears, Figure 5 , Figure 7 , Figure 11 axis are not clearly readable.

Response：We have made correction according to the Reviewer’s comments.We provide all the high-definition vector images

9.It would be more transparent if authors can provide their R scripts used for the analysis for quick check.

Response：We have made correction according to the Reviewer’s comments. We provided the original transcriptome data, important data for this study, and the R script used to build WGCNA, based on the reviewers' recommendations.

WGCNA: usage and download of an R package for weighted correlation network analysis(https://horvath.genetics.ucla.edu/html/CoexpressionNetwork/Rpackages/WGCNA/index.html#manualInstall).

Reviewer #2:

1. Authors say that data is available without restrictions but I did not see a link to the data.

Response：We have made correction according to the Reviewer’s comments. We provided the original transcriptome data, complete data for this study, and the R script used to build WGCNA, based on the reviewers' recommendations.

2.The English and grammar is not upto the scientific standards. Needs a lot of correction.

Response：We are sorry that our English writing level has troubled the reviewers. We have submitted our manuscripts to international science editing(http://www.internationalscienceediting.com) for polishing as suggested by the reviewers.

3.Figure legends are not in a standard format

Response：We have made correction according to the Reviewer’s comments.We provide all the high-definition vector images

4.Authors do not describe the sequencing data analysis. Which tools were used in processing and normalization etc. No description of any statistics that was performed and should have been performed the assess the gene expression levels of the gene of interest from the sequencing data. The raw data should be made available so people can reproduce the results.

Response：We are very grateful to the reviewers for this error, and we have added the functions and methods used in each operation step in the materials and methods.We also provided the original transcriptome data, important data for this study, and the R script used to build WGCNA, based on the reviewers' recommendations.

---

## [Decision Letter · Decision Letter 1]

23 Nov 2020

Using WGCNA（weighted gene co-expression network analysis)  to identify the hub genes of skin hair follicle development in fetus stage of Inner Mongolia Cashmere goat

PONE-D-20-17381R1

Dear Dr. Yanjun Zhang,

We’re pleased to inform you that your manuscript has been judged scientifically suitable for publication and will be formally accepted for publication once it meets all outstanding technical requirements.

Kind regards,

Naresh Doni Jayavelu, Ph.D

Academic Editor

PLOS ONE

Additional Editor Comments (optional):

Reviewers' comments:

Reviewer's Responses to Questions

**Comments to the Author**

1. If the authors have adequately addressed your comments raised in a previous round of review and you feel that this manuscript is now acceptable for publication, you may indicate that here to bypass the “Comments to the Author” section, enter your conflict of interest statement in the “Confidential to Editor” section, and submit your "Accept" recommendation.

Reviewer #1: All comments have been addressed

Reviewer #2: All comments have been addressed

2. Is the manuscript technically sound, and do the data support the conclusions?

Reviewer #1: Partly

Reviewer #2: Yes

3. Has the statistical analysis been performed appropriately and rigorously? 

Reviewer #1: I Don't Know

Reviewer #2: Yes

4. Have the authors made all data underlying the findings in their manuscript fully available?

Reviewer #1: Yes

Reviewer #2: Yes

5. Is the manuscript presented in an intelligible fashion and written in standard English?

Reviewer #1: Yes

Reviewer #2: Yes

6. Review Comments to the Author

Reviewer #1: Thanks for incorporating previous suggestion. Now manuscript looks much better. I dont have any major comments except for few typo for e.g. wayne map or venn diagram ?

Reviewer #2: My queries have been answered. I think the authors have tried to improve the English and re written the introduction.

7. PLOS authors have the option to publish the peer review history of their article (what does this mean?). If published, this will include your full peer review and any attached files.

Reviewer #1: No

Reviewer #2: No

---

## [Editor Report · Acceptance letter]

9 Dec 2020

PONE-D-20-17381R1 

Using WGCNA（weighted gene co-expression network analysis)  to identify the hub genes of skin hair follicle development in fetus stage of Inner Mongolia Cashmere goat 

Dear Dr. Zhang:

I'm pleased to inform you that your manuscript has been deemed suitable for publication in PLOS ONE. Congratulations! Your manuscript is now with our production department. 

Kind regards, 

on behalf of

Dr. Naresh Doni Jayavelu 

Academic Editor

PLOS ONE